# Programmatic Cost-Effectiveness of a Second-Time Visit to Detect New Tuberculosis and Diabetes Mellitus in TB Contact Tracing in Myanmar

**DOI:** 10.3390/ijerph192316090

**Published:** 2022-12-01

**Authors:** Nyi Nyi Zayar, Rassamee Chotipanvithayakul, Kyaw Ko Ko Htet, Virasakdi Chongsuvivatwong

**Affiliations:** Department of Epidemiology, Faculty of Medicine, Prince of Songkla University, Hat Yai, Songkhla 90110, Thailand

**Keywords:** screening, tuberculosis, diabetes mellitus, cost-effectiveness, compliance

## Abstract

Background: Integration of diabetes mellitus screening in home visits for contact tracing for tuberculosis could identify hidden patients with either tuberculosis or diabetes mellitus. However, poor compliance to the first home screening has been reported. A second time visit not only increases screening compliance but also the cost. This study aimed to determine if an additional second time visit was cost effective based on the health system perspective of the tuberculosis contact tracing program in Myanmar. Methods: This cross-sectional study was based on usual contact tracing activity in the Yangon Region, Myanmar, from April to December 2018 with integration of diabetes mellitus screening and an additional home visit to take blood glucose tests along with repeated health education and counseling to stress the need for a chest X-ray. New tuberculosis and diabetes mellitus cases detected were the main outcome variables. Programmatic operational costs were calculated based on a standardized framework for cost evaluation on tuberculosis screening. The effectiveness of an additional home visit was estimated using disability-adjusted life years averted. The willingness to pay threshold was taken as 1250.00 US dollars gross domestic product per capita of the country. Results: Single and additional home visits could lead to 42.5% and 65.0% full compliance and 27.2 and 9.3 additional years of disability-adjusted life years averted, respectively. The respective base costs and additional costs were 3280.95 US dollars and 1989.02 US dollars. The programmatic costs for an extra unit of disability-adjusted life years averted was 213.87 US dollars, which was lower than the willingness to pay threshold. Conclusions: From the programmatic perspective, conducting the second time visit for tuberculosis contact tracing integrated with diabetes mellitus screening was found to be cost effective.

## 1. Introduction

Globally, tuberculosis (TB) is the most prevalent cause of mortality from a single infectious pathogen, which caused 1.2 million deaths in 2019 [1]. High blood glucose is the third highest risk factor of premature mortality [2], and diabetes mellitus (DM) attributed to 4.2 million deaths also in 2019 [3]. Moreover, the burden of hidden TB and DM cases has been on the rise. It was estimated that 2.9 million (29%) cases of TB and almost 30–50% of those with diabetes remained undiagnosed [3,4].

Myanmar is an endemic area for both TB and DM. The prevalence of DM in Myanmar was reported as 10.5% [5], which was higher than the regional and global estimates. In addition, Myanmar was among the 30 countries with the highest TB burden and the ninth among countries with the highest incidence of TB–DM comorbidity [6]. Patients with TB–DM comorbidity are likely to have poor treatment outcomes [7], which calls for urgent appropriate health care provisions.

Myanmar has responded to the high TB burden since 2015 by implementing household contact tracing, which is the most cost-effective program for active case findings of undiagnosed TB patients in the community [8,9]. Subsequently, the package of essential non-communicable disease intervention developed by the World Health Organization (WHO) offers an opportunity to screen for DM in the primary healthcare clinic setting, which was started in some townships in 2018 [10]. However, this program may miss those with DM who do not visit a clinic. Later in the same year of 2018, Myanmar started a bidirectional screening program to screen for DM among TB patients [11]. Integration of diabetic screening in routine TB contact tracing can increase the detection of undiagnosed TB as well as undiagnosed DM among household contacts of TB patients.

A challenge for the diagnosis of DM is to perform complete blood testing two times as required by the American Diabetes Association [12]. A home-based screening program for DM in Kenya that reported only 23% of those who had a first positive test result during the home visit went to a healthcare facility to complete the second blood test to confirm the DM diagnosis [13]. Similarly, poor compliance is also a major barrier in TB contact tracing. Previous studies reported only 33.7% in Ethiopia and 54.0% in Thailand brought their household contacts to TB clinics [14,15]. Repeated home visits to encourage TB screening could increase compliance as well as the cost [16]. Based on a simulation model, the operational costs of these multiple home visits was cost effective [8]. However, it has never been examined using real data collected from an actual program by integrating both TB and DM screening. Therefore, this study aimed to identify the level of compliance to TB and DM screening after a second time visit and determine the incremental cost-effectiveness of a second time visit over a single home visit related to contact tracing using data collected from a primary research study.

## 2. Materials and Methods

### 2.1. Study Design and Population

A cross-sectional study on community-based contact tracing with a second time visit was conducted in North Okkalapa and Insein townships in the Yangon Region from April to December 2018. Newly diagnosed smear positive index TB patients who registered in the township TB clinics were identified. The household contacts who lived in the same household with TB patients for at least three months were included in our study.

The sample size of household TB contact tracing was estimated based on 4.5% prevalence of TB from household contact tracing in low- and middle-income countries [17]. With 3% precision and a design effect of 1.5 with a 10% non-response rate, at least 306 household contacts were needed. The sample size of household diabetic screening was estimated based on 12.1% of DM in the population aged ≥25 years in Yangon [18], 5% precision, 1.5 design effect, and a 10% non-response rate. Finally, at least 271 household contacts were required.

### 2.2. Study Setting

The study area has a high number of TB and DM cases. The TB case notification rates in North Okkalapa and Insein townships were 387 and 248 per 100,000 population, respectively [19]. The Yangon Region had the highest DM prevalence of 12.1% in Myanmar [18].

Routine TB contact tracing in Yangon Region was conducted by the Basic Health Staff under supervision of the National Tuberculosis Programme. The Basic Health Staff performed contact tracing usually one time within one month after the index TB patient initiated the anti-TB treatment [20]. Initial screening of TB among household contacts of TB patients was performed by detecting TB signs and symptoms during the home visit. Persons with screen positive were referred to the nearest township TB clinic to undergo sputum smear tests and a chest X-ray (CXR) to confirm TB. The GeneXpert test was performed to detect multidrug resistant TB among newly diagnosed TB patients. DM screening was also performed in newly diagnosed index TB patients according to WHO recommendations. However, screening for DM was not routinely performed in conventional contact tracing.

### 2.3. Contact Tracing in the Study

Screening of TB in household contacts was performed using CXR at the township TB clinic according to WHO recommendations [21]. In addition, we performed DM screening in persons ≥ 25 years old by performing random blood glucose (RBG) tests during the first home visit and fasting blood glucose (FBG) tests in the township TB clinic together with CXR examinations. For those who did not come to the TB clinic within two weeks after the first home visit, we revisited their home to perform the FBG test and conduct repeat health education and counseling to stress the need for a CXR examination. The differences between the routine contact tracing in Myanmar, WHO recommendations, and our study are shown in Appendix A.

### 2.4. Data Collection

A list of newly diagnosed sputum-positive TB patients ≥ 25 years old was obtained from the township TB clinic registers, and those patients with at least one household contact were invited to join the study. Contact tracing was performed in all household contacts of the index TB patients. The contact tracing data were collected according to steps described in the last column of Appendix A. Human immunodeficiency virus testing was also performed in patients diagnosed with TB after counseling. In summary, our protocol followed the WHO recommendations but added RBG and FBG tests for household contacts ≥25 years old and added a second time visit to take the FBG test and repeat the health education and counselling session to stress the need for a CXR examination.

Instead of using a sputum smear, our diagnosis of TB was made using the GeneXpert positive results or a clinical diagnosis based on CXR findings after the antibiotic trial test (Appendix A). For any single positive test result of RBG or FBG, the test was repeated on a separate day. Newly diagnosed DM was defined as subjects with RBG ≥ 200 mg/dL and FBG ≥ 126 mg/dL, or RBG results that were ≥200 mg/dL on two occasions on separate days, or FBG ≥ 126 mg/dL on two occasions on separate days [12].

All positive TB patients were registered at the township TB clinics and received treatment according to the National TB treatment guidelines. All positive DM patients were referred to DM clinics for further examinations for complications and appropriate treatment.

### 2.5. Evaluation of Healthcare Programmatic Operational Costs

Our study computed the operational costs of a healthcare program on screening of TB and DM in contact tracing. Cost estimation was performed based on the standardized framework for cost evaluation of a TB screening program [22]. The estimated costs were obtained from primary data collection and the procurement section of the National TB Control Programme under the Global Fund Project in 2016 [23]. The total costs for single home contact tracing visits and second time contact tracing visits were calculated.

#### 2.5.1. Human Resource Costs

The costs of human resources were calculated by multiplying the working hours spent on individual household contact from the wages per hour. The monetary value of local currency was converted to US dollars (USD) (1 USD = 1328 Myanmar Kyat [MMK] in 2018) [24]. As a result, 1.03 USD per hour was used for volunteers and basic health staff including the township TB coordinator, 1.08 USD per hour for a medical technologist, 1.23 USD per hour for a CXR technician, and 1.50 USD per hour for a medical doctor and microbiologist.

#### 2.5.2. Capital Costs

The annualized value of capital costs for buildings and medical devices used were calculated by dividing the current value of the buildings or items by the annualizing factor, which was computed based on the useful life of the assets at a 3% discount rate [25]. According to Myanmar Ministry of Finance [26], the useful life of a building was estimated as 50 years and hospital equipment as 10 years. The annualization factor was calculated using the following formula:$$[{\left(1+r\right)}^{n}-1]/[r{\left(1+r\right)}^{n}]$$
where *r* = 3% discount rate and *n* = total years of useful life. Calculations of the capital costs per test are shown in Appendix A.

#### 2.5.3. Recurrent Costs

Travel costs for home visits by healthcare volunteers were calculated based on the cost of using public transportation per household. The total travel cost for home contact tracing visits was divided by the total number of household contacts involved in the study to arrive at the average travel cost per household contact. The cost of telephone calls was calculated by dividing the total hours of phone calls to households divided by the total number of household contacts. The cost of materials used for TB screening was estimated from the market-based costs obtained from the procurement section of the National TB Control Programme under the Global Fund Project conducted in 2016. The average cost of materials was calculated as the cost per individual by dividing the total cost of all materials by the total number of household contacts who underwent the test. All values of items in Myanmar Kyat were converted to USD (1.00 USD = 1328 MMK).

#### 2.5.4. Overhead Costs

Overhead costs included maintenance, calibration, and annual electricity consumption (kW/h) charges for the CXR machines and GeneXpert tests. The costs per individual testing was calculated by dividing the annual costs of each category by the total number of tests taken in 2018. Calculations of the overhead cost per testing are shown in Appendix A.

#### 2.5.5. Compliance to Screening

Full compliance is defined as compliance to both TB and DM screening after a home visit among household contacts ≥25 years old or compliance to TB screening after a home visit among household contacts <25 years old. No compliance is defined as no compliance to both TB and DM screening after a home visit among household contacts ≥25 years old or no compliance to TB screening after a home visit among household contacts <25 years old. Partial compliance means the household contacts ≥25 years old who complied either with the TB screening or the DM screening.

#### 2.5.6. Disability-Adjusted Life Year Averted

The effectiveness of special contact tracing in our study was quantified using the disability-adjusted life years (DALY) averted defined as the number of years lived with full health due to early detection of a new case [27]. However, since there is limited local information available to calculate DALY averted for TB and DM screening, the parameters were estimated from previous studies performed in Cambodia where socioeconomic background, TB, and DM patterns were similar to Myanmar [28]. The reference values used in the Cambodian study to calculate DALY averted were also applied in other previous studies following the WHO guide to cost-effectiveness analyses [29,30]. Therefore, DALY averted value per one newly detected TB case was 1.8 years [9]. This study was cited by studies conducted in Pakistan and Ethiopia [31,32]. DALY averted value per one newly detected DM case was referenced from a simulation study for low- and middle-income countries [33], which was 0.038 years. These two values were calculated ignoring the age and sex of the detected subjects.

#### 2.5.7. Decision Tree

The decision tree with probabilities of detecting new TB and DM cases and their results in DALY averted based on three levels of compliances were developed according to findings in our study tree (Appendix A). The decision tree shows the probabilities of identifying new TB and DM cases from the first- and second-time visits. Missing data in any of the screening tests resulted in zero DALY averted.

### 2.6. Data Analysis

Epidata version 3.1 [34] was used to enter the collected data, and the data analysis was performed using R version 4.1.2 [35]. Information for percentage of compliance to TB and DM screening was interpreted along with the number of cases detected. The level of compliance to screening is presented in frequency and percentage. The incremental cost-effectiveness ratio (ICER) was determined using the following formula:$$\mathrm{ICER}=\frac{\mathrm{Cos}\mathrm{ts}\;\mathrm{of}\;\mathrm{two}\;\mathrm{visits}-\mathrm{Cos}t\;\mathrm{of}\;\mathrm{first}\;\mathrm{visit}\;}{\mathrm{DALY}\;\mathrm{averted}\;\mathrm{by}\;\mathrm{two}\;\mathrm{visits}-\mathrm{DALY}\;\mathrm{averted}\;\mathrm{by}\;\mathrm{first}\;\mathrm{visit}}$$

The second time visit was considered cost effective if the cost per DALY averted was below the value of one GDP per capita (1250.00 USD) [29,36].

#### 2.6.1. Probabilistic Sensitivity Analysis

The above analysis was based on a single value for each parameter, which in fact had some uncertainty [37]. Probabilistic sensitivity analysis [38] was used to allow random changes in the parameters listed in Appendix A. Beta and gamma distributions were used to model the uncertainty in conducting sensitivity analysis following most of the previous studies [39,40]. A beta distribution was used to estimate the 95% confidence interval for the probability of continuous random variables with range of 0 to 1, which meets compliance to screening in our study. Gamma distribution was used to estimate the 95% confidence interval for parameters with non-negative values with an interval of 0 to positive infinity, which are costs and DALY averted parameters in the study [41]. Parameters for Monte Carlo simulation [42] included in the decision tree are shown in Appendix A. As a result of 10,000 simulations, we obtained 10,000 pairs of costs and effectiveness values. They were plotted on a plane with the DALY averted value attained after a second time visit on the *x*-axis and the incremental cost on the *y*-axis. The threshold line joining the origin and the coordinate point of DALY averted and cost was the value of ICER. The proportions of points that fall to the right and below the threshold line indicated the probability of cost-effectiveness. R software was used to generate the figure.

#### 2.6.2. One-Way Sensitivity Analysis

The ICER value strongly depended on the level of percentage of full compliance from the first home visit, which was uncertain. To focus on this uncertainty in the study, one-way sensitivity analysis was performed to vary the level of this compliance while keeping other parameters unchanged. The relationship between the ICER value and the percentage of full compliance was then visualized in a two-dimensional line graph. The figure was generated by using R software.

## 3. Results

### 3.1. Compliance to Screening

Among 553 household contacts recruited, 33 with a known history of DM and one known case for TB were excluded. Therefore, 519 household contacts were included in the analysis.

Out of the first home visit, 221 (42.5%) patients came to the township TB clinic and fully complied to both the TB and DM investigations (Table 1). Subsequently, second time visits were conducted to persuade the remaining persons who were untested or had incomplete testing to fully comply with the investigations. The second time visit resulted in an additional 178 household contacts who fully complied with the required investigations, which resulted in a total of 399 participants that increased the compliance to 76.9% of the target.

### 3.2. Operational Costs and DALY Averted

The total operational costs after the first- and second-time visits were 3280.95 USD and 1989.02 USD, respectively (Table 1). The average costs per household contact were 6.32 USD and 6.83 USD after the first- and second-time visits, respectively. Appendix A compares the complete figures of operational costs of contact tracing after the first home visit, second-time visit, and two home visits.

The yield to identify new TB cases was higher in the first visit (15 cases) compared to the second (5 cases). The yield in detecting new DM had the opposite direction of difference (4 vs. 9, respectively). Multiplying these numbers with the estimated DALY averted per case detected and dividing the total costs of each visit with these numbers, the first home visit cost was 120.62 USD per DALY averted. The second visit had an ICER of 213.87 USD per DALY averted.

### 3.3. Cost-Effectiveness Analysis of All Contacts Using Monte Carlo Simulation

Figure 1 shows the incremental cost-effectiveness plane, in which the *x*-axis reflects the DALY averted value attained after the second-time visit, and the *y*-axis represents the incremental costs of the second-time visit. The horizontal axis divides the incremental cost as positive above and negative below its origin (*x*-axis = 0), while the vertical axis divides the DALY averted value into the positive to the right and the negative to the left of its origin (*y*-axis = 0). Three dashed lines pass the origin with different slopes. Line “A” has a slope of 1250.00 USD per DALY averted. This is the public investment threshold of the willingness to pay (WTP). All of the area below and on the right side of this line are lower than the WTP threshold or have a good cost-effectiveness ratio. Line “B” has the maximum slope among all 10,000 simulated dots. This denotes that the highest ICER value from the simulation was 275.00 USD per DALY averted. Line “C” passes the median ICER value of 213.90 USD per DALY averted. In summary, the median ICER value was 17% of the WTP. All simulation values were below the WTP threshold.

### 3.4. One-Way Sensitivity Analysis for Different Percentages of Full Compliance after the First Home Visit to Assess the Cost-Effectiveness of a Second-Time Visit

Figure 2 depicts the one-way sensitivity analysis showing the change in ICER value of the second-time visit with changing percentages of full compliance to screening after the first home visit. The *x*-axis represents the percentage of full compliance to the first home visit, and the *y*-axis reflects the ICER values. The ICER values ranged from 129.90 USD at 0% of full compliance to 1247.90 USD at 56.7% of full compliance before it crossed the WTP threshold of 1250.00 USD. Therefore, the second-time visit was cost effective when full compliance to screening after first home visit was below 56.7%.

## 4. Discussion

This paper highlighted that the second-time visit for TB and DM, which is rarely conducted in a routine healthcare program, is cost effective It could increase full compliance to TB and DM screening from 42.5% to 76.9% after the second-time visit. Furthermore, contact tracing with second-time visits could avert an additional 9.3 years of DALY. The extra programmatic operational unit cost for one extra unit of DALY averted was 213.87 USD. Upon setting the WTP threshold of one GDP per capita (1250.00 USD) per DALY averted, the second-time visit attained a 100% probability of cost-effectiveness. However, one-way sensitivity analysis showed that conducting second-time visits was beneficial and was cost effective especially since full compliance to screening after the first home visit was lower than 56.7%.

According to WHO guidelines on contact tracing, home visits are recommended in index TB patients to perform interviews and ensure referral of household contacts for clinical evaluation [43]. Studies in low- and middle-income countries, however, showed that compliance was poor, which ranged from 33.7% to 42.9% for TB screening [14,44]. Additionally in our study, only 42.5% of household contacts fully complied to screening after the first home visit. However, after the second-time visit, compliance to screening increased to 76.9%, which was likely related to the FBG test and repeated health education with counselling. During the second-time visit, a CXR examination was strongly suggested, which proved to be effective since the compliance rates were below 56.7% after the first contact tracing.

This is the first study to evaluate the cost-effectiveness using real data of repeating a home visit in TB contact tracing integrated with DM screening. The DALY averted in our study reflects the effectiveness of both TB and DM screening among household contacts of TB patients. An additional 9.3 DALY was averted after the second-time visit. A major portion of the additional DALY averted was attributed by newly detected TB patients due to the relatively increased DALY value of newly detected TB patients compared to newly detected DM, (1.8 vs. 0.038) [9,33]. The additional cost to avert one DALY was 213.87 USD after the second-time visit, which was about 18% of GDP per capita of the country [36]. Therefore, spending the additional 213.87 USD for the second-time visit to improve screening of household contacts in contact tracing could return 1250.00 USD GDP per year to the country.

The average cost of contact tracing per contact in our study was 6.32 USD for the first home visit and a 6.83 USD increment for the second-time visit. Both of these costs were higher than the 4.90 USD for a single home visit in a study conducted three years previously in the Mandalay Region [23]. The higher average cost in our study might be due to inclusion of the training costs for healthcare volunteers, costs for DM screening, and the travel costs, which would be different due to the different study settings. However, we did not include the cost for the loss of productivity of the household contacts when they visited the TB clinics.

The extra cost for one unit of DALY averted by screening of TB and DM after a single home visit was 120.62 USD. The incremental costs of an active case finding in previous studies conducted in Pakistan [45], Cambodia [9], and Peru [8] were 238.00 to 1811.00 USD. All of the studies were found to be cost effective compared to their respective WTP threshold levels. This may suggest that an active case finding was found to be more cost effective than a passive case finding.

In our study, DM screening was performed among household contacts ≥25 years old regardless of having the risk of a non-communicable disease. Therefore, the results can be considered as a population-based screening study in the household contacts of TB patients. A previous study conducted in Brazil to compute the cost-effectiveness of population-based screening of DM reported that the intervention was not cost effective [46]. However, another study performed in Bhutan for the cost-effectiveness of universal screening of DM represented good value for money compared to no screening [47]. The effectiveness of a screening program is largely influenced by the prevalence of disease. The prevalence of DM in Brazil (4.4%) [48] was relatively low compared with Bhutan (7.7%) [49]. The prevalence of DM in the Yangon Region (12.2%) [18], which was significantly higher than the prevalence in Brazil and Bhutan, subsequently contributed to the cost-effectiveness of screening for TB and DM in our study.

In the sensitivity analysis, conducting the second-time visit would not be cost effective if more than 56.7% of household contacts fully complied to screening after the first home visit. A previous study in the Mandalay Region arranged for patient transportation to undergo CXR examinations in the township TB clinics to improve compliance with the screening [50]. In our study, visiting the TB clinic to take the FBG test in the early morning is also the major reason for poor compliance. Screening of DM using the glycated hemoglobin (HbA1c) test is one of the alternatives without fasting, although the cost is higher than the FBG test [12]. Therefore, the provision of transportation and screening of DM using the HbA1c test would likely increase compliance to screening of TB and DM from the first home visit.

The study reflected the urban area where burden of both TB and DM was high due to crowded living conditions. Our study findings are not only useful for Myanmar but also for other low- and middle-income countries (LMICs), such as India and Ethiopia, as well as, some upper middle-income countries, such as China [14,51,52]. Due to limitations of the study design and the availability of data, we could not include the societal and treatment costs. It would be a challenge to accurately measure the lifetime treatment costs of DM. The value of DALY averted was adapted from other countries due to a lack of such data in Myanmar.

The strength of the study were that compared with the simulation-based cost-effectiveness study in Peru [8], our findings gave strong evidence that a single visit to a TB household for contact tracing is inadequate and a repeated visit can be cost effective especially in the LMICs with poor compliance to screening [16,53]. Additionally, the study design followed the approach of WHO-recommended patient-centered care for TB that is an essential component in the End TB strategy [54].

## 5. Conclusions

The second-time visit was cost effective in high TB- and DM-burdened areas as evidenced by higher compliance to TB and DM screening after the second visit.

## Figures and Tables

**Figure 1 ijerph-19-16090-f001:**
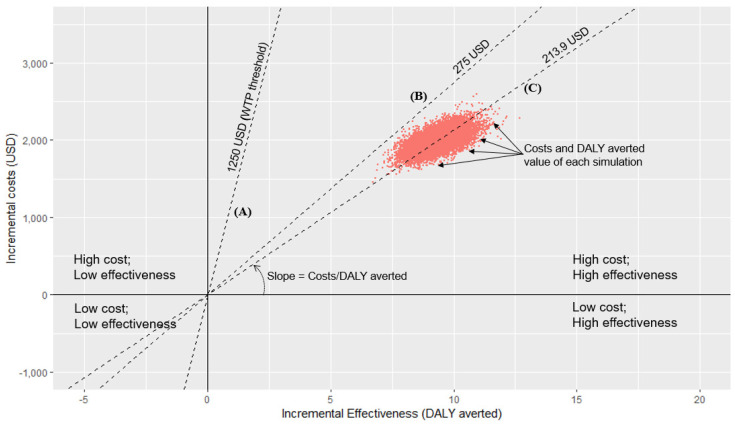
Incremental cost-effectiveness plane showing incremental cost-effectiveness ratios of second-time visits using Monte Carlo simulation. WTP: willingness to pay. *Note:* Line (A)—Slope for WTP threshold (1250.00 USD) per DALY averted; Line (B)—The maximum slope among all 10,000 simulated dots; Line (C)—Slope passes the median ICER value of 213.90 USD per DALY averted.

**Figure 2 ijerph-19-16090-f002:**
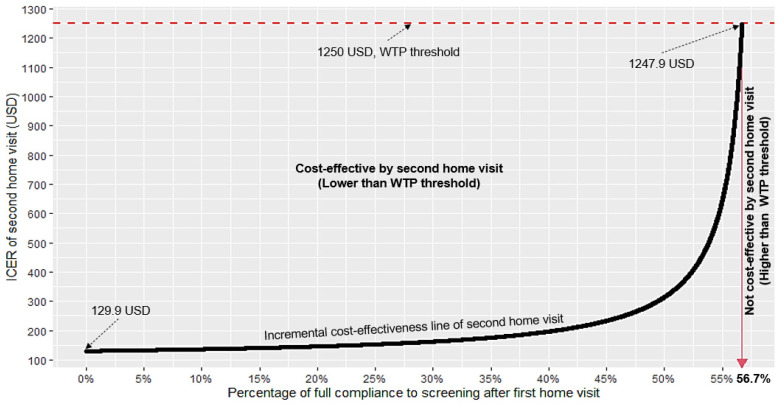
Graph showing ICER value changes with different percentages of full compliance from the first home visit. ICER: incremental cost-effectiveness ratio; WTP: willingness to pay.

**Table 1 ijerph-19-16090-t001:** Comparison of compliance, DALY averted, and operational costs between the first- and second-time visits.

	First-Time Visit *n* = 519	Second-Time Visit *n* = 291	Two-Time Visits *n* = 519
Level of compliance			
Full compliance	221 (42.5)	178 (65.0)	399 (76.9)
Compliance to TB screening	231 (44.5)	184 (67.2)	415 (79.9)
Compliance to DM screening	127 (40.6)	167 (89.8)	294 (93.9)
New TB cases detected	15	5	20
DALY averted *	27.0	9.0	36.0
New DM case detected	4	9	13
DALY averted *	0.2	0.3	0.5
Total DALY averted	27.2	9.3	36.5
Operational costs (USD)			
Total costs	3280.95	1989.02	5269.97
Costs per household contact	6.32	6.83	10.15
Costs per DALY averted	120.62	213.87 ***	144.38

Data are presented as *n* or *n* (%) unless otherwise indicated. DALY: disability-adjusted life years; * DALY averted = 1.8 and 0.038 DALY averted per one new TB and DM detected, respectively. *** ICER incremental cost-effectiveness ratio of second-time visit.

## Data Availability

The datasets supporting the conclusion of this article are included within the article and its additional files.

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
