# Peer review of "Programmatic Cost-Effectiveness of a Second-Time Visit to Detect New Tuberculosis and Diabetes Mellitus in TB Contact Tracing in Myanmar"

_ijerph, 2022, doi:10.3390/ijerph192316090_

Round 1
Reviewer 1 Report
The paper in its current form has the following MAJOR drawbacks from my viewpoint:
1- The scope of the paper is too narrow and the contribution of the paper is limited for publication. This is a modest extension of existing work with marginal contribution.
2- The title doesn't reflect the contribution of the paper. The title must contain the region (Myanmar) and time (April-December 2018) of this study. The abstract also must be revised.
3- The number of cases (519) and time period (April-December 2018) make the study very limited.
4- In page 4, you mentioned that the Beta and Gamma distributions will be used without any scientific justification. Therefore, the reader may ask: Why these two distributions will be used?
5- The writing and the presentation need significant improvements.
In general, the contribution from this paper in its current form is very marginal, thus not significant and sufficient to be considered as a full length research paper.
Author Response
Response to Reviewer 1 Comments
Dear Editors and Reviewers,
We are kindly submitting the revised version of our manuscript “Programmatic Cost -Effectiveness of a Second Time Home Visit to Detect New Tuberculosis and Diabetes Mellitus in
TB Contact Tracing in Myanmar from April to December 2018” (Manuscript ID: ijerph-2019883)
Reviewers (1)
Comments and Suggestions for Authors
Revision of the manuscript has been done using Track Changes in MS word.
The paper in its current form has the following MAJOR drawbacks from my viewpoint:
Point 1: The scope of the paper is too narrow and the contribution of the paper is limited for publication. This is a modest extension of existing work with marginal contribution.
Response 1: Both TB and DM are major endemic diseases in both low- and middle-income and upper middle-income countries. WHO recommends bidirectional screening of TB and DM in healthcare facilities. On the other hand, it was estimated that almost 20% of TB patients and 30–50% of DM patients were hidden in the community. In the study, screening of TB and DM was done in routine contact tracing among household contacts of TB patients to explore undiagnosed TB or DM patients in the community. However, poor compliance to screening is one of the major barriers; therefore, there is a need for effective intervention, which subsequently increases the costs. This paper highlights the cost-effectiveness of conducting second home visits in screening of TB and DM among household contacts of TB patients. These facts were mentioned in the background of the article.
Point 2: The title doesn't reflect the contribution of the paper. The title must contain the region (Myanmar) and time (April-December 2018) of this study. The abstract also must be revised.
Response 2: In title, we included “in Myanmar”. In the abstract, we revised region and time accordingly.
Please see (L5) and (L21)
Point 3: The number of cases (519) and time period (April-December 2018) make the study very limited.
Response 3: We have added the sample size calculation
Please see (L103 - 109)
Point 4: In page 4, you mentioned that the Beta and Gamma distributions will be used without any scientific justification. Therefore, the reader may ask: Why these two distributions will be used?
Response 4: In the study, the unit costs and DALY averted values were obtained from external sources which may lead to parameter uncertainty which needs assumption of the probability distribution. Beta and gamma probabilistic distributions were used as most of the previous studies did. We also added more details on their use in the manuscript.
Please see (L248 - 255)
Point 5: The writing and the presentation need significant improvements.
Response 5: The manuscript has been further copy edited by a native English person with 11 years of medical editing experience, working at the international affairs office, Faculty of Medicine at Prince of Songkla University, Thailand.
Point 6: In general, the contribution from this paper in its current form is very marginal, thus not significant and sufficient to be considered as a full-length research paper.
Response 6: Our manuscript will be beneficial for policy decision makers to consider the appropriate intervention for integrating both TB and DM screening in the community as an alternative option, rather than screening only at the health facilities where compliance was one of the barriers. Strengthening the existing health care of TB contact tracing by integrating both TB and DM screening at the community level will improve early detection of TB and DM cases which have a high burden of morbidity and mortality. Therefore, this approach, “integrated and patient centered care and prevention of TB and its comorbid DM” is one of the three main pillars and components and will contribute to the End TB strategy.

Reviewer 2 Report
"Second home visit"-- advised to change as Second time Visit / Repeated Visit for second time.
The English fluency is less , kindly make more fluent.
Author Response
Response to Reviewer 2 Comments
Dear Editors and Reviewers,
We are kindly submitting the revised version of our manuscript “Programmatic Cost -Effectiveness of a Second Time Home Visit to Detect New Tuberculosis and Diabetes Mellitus in
TB Contact Tracing in Myanmar from April to December 2018” (Manuscript ID: ijerph-2019883)
Reviewer (2)
Comments and Suggestions for Authors
Point 1: "Second home visit"-- advised to change as Second time Visit / Repeated Visit for second time.
Response 1: Revised accordingly in the manuscript using Track Changes.
Point 2: The English fluency is less, kindly make more fluent.
Response 2: The manuscript has been further copy edited by a native English person with 11 years of medical editing experience, working at the international affairs office, Faculty of Medicine at Prince of Songkla University, Thailand.

Reviewer 3 Report
Minor revisions on some concerned areas are suggested

Author Response
Response to Reviewer 3 Comments
Dear Editors and Reviewers,
We are kindly submitting the revised version of our manuscript “Programmatic Cost -Effectiveness of a Second Time Home Visit to Detect New Tuberculosis and Diabetes Mellitus in
TB Contact Tracing in Myanmar from April to December 2018” (Manuscript ID: ijerph-2019883)
Reviewer (3)
The paper attempts to study the cost effectiveness of a second home visit to detect new tuberculosis and diabetes mellitus in TB contact tracing. It is a good attempt so far as the social issue and sustainable development are concerned. However there are some left out issues, which I think, should have been addressed, and thus the paper requires minor revision.
Response: Thanks reviewer for your valuable suggestions.
Revision of the manuscript has been done using Track Changes in MS word.
Point 1: The Abstract should clearly present the objective/s (e.g., The study aimed to determine the cost effectiveness of an additional home visit from the health system perspective of what?) and should not contain any abbreviated terms. It is suggested thus to revise the abstract.
Response 1: We revised objectives as “The study aimed to determine if an additional second time visit was cost effective based on the health system perspective of the tuberculosis contact tracing program in Myanmar”. Please see (L 17 – 19)
Abbreviated terms were replaced with full words in the abstract.
Point 2: In the Introduction part, the authors have given sufficient background information on the said topic but they did not identify the gaps in the literature. It is thus suggested to write a paragraph on the Research Gap and frame the objectives accordingly.
Response 2: We revised the paragraph on the Research Gap and framed the objectives accordingly.
A single home visit for DM screening had poor compliance. In the meantime, repeated home visits were done to encourage household contacts of TB patients to visit a health facility for TB screening could increase the compliance as well as the cost. Based on a simulation model, the operation of these multiple home visits was cost effective. However, it has never been examined using real data collection from an actual program by integrating both DM and TB screening. Therefore, this study aimed to identify the level of compliance to TB and DM screening after a second time visit, and to determine the incremental cost -effectiveness of a second time home visit over a single home visit related to contact tracing using data collected from a primary research study.
Please see (L68 – 95)
Point 3: In the section on Disability‐Adjusted Life Year Averted, the authors mentioned non availability of the local information for the calculation of DALY they have used the Cambodian reference. It is good but there is no reason why they chose the reference. Were there no alternatives? If no, try to cite some other works where this particular reference is used.
Response 3: We added reasons to take the reference from the Cambodian study and cited other works to this paper.
Please see (L 214 – 218)
Point 4: The figures and tables do not have sources. It is suggested to add.
Response 4: The data shown in the tables and figures were generated by our own analysis using R software.
R software was used to generate the figure. Please see (L 271)
Point 5: The results of the study could not be robust unless the comparative analyses with other studies are made. It is thus suggested to make comparisons of the related works to identify itself as the significant contributors to the literature.
Response 5: This is the first study to determine the cost-effectiveness of conducting second time visits in routine TB contact tracing integrated with DM screening. There has never been any studies looking at these issues.

Round 2
Reviewer 1 Report
As I mentioned in my first-round report: The scope of the paper is too narrow, the contribution of the paper is limited, and the time period for the study and its location are also limited. Therefore, the conclusion cannot be extended to more general period or locations. Then, the paper in its current form is not interesting enough for reader.
